# Enzymatic Hydrolysis of Rutin: Evaluation of Kinetic Parameters and Anti-Proliferative, Mutagenic and Anti-Mutagenic Effects

**DOI:** 10.3390/life13020549

**Published:** 2023-02-16

**Authors:** Mariana Alves Sobreiro, Adriana Della Torre, Maria Elisa Melo Branco de Araújo, Paula Renata Bueno Campos Canella, João Ernesto de Carvalho, Patrícia de Oliveira Carvalho, Ana Lucia Tasca Gois Ruiz

**Affiliations:** 1Health Sciences Postgraduate Program, Universidade São Francisco (USF), Bragança Paulista 12916-900, SP, Brazil; 2Laboratory of Phytochemistry and Experimental Pharmacology and Toxicology (LAFTEX), Faculty of Pharmaceutical Sciences, University of Campinas, 200 Candido Portinari Street, Campinas 13083-871, SP, Brazil

**Keywords:** quercetin-3-O-glucoside, hesperidinase, genotoxicity test

## Abstract

The bioavailability of glucoside flavonoids is influenced by the nature of the sugar, glucosides being absorbed faster than rhamnoglucosides, for example. One strategy to enhance the bioavailability is enzymatic hydrolysis. In this study, some kinetic parameters of hesperidinase-mediated hydrolysis of rutin were evaluated using an UHPLC/QTOF-MS^E^ analysis of the products of a bioconversion reaction. The resulting hydrolyzed rutins (after 4, 8 and 12 h of reaction) were submitted to anti-proliferative and Cytokinesis-Block Micronucleus (CBMN) assays in CHO-K1 cells. In the hesperidinase-mediated hydrolysis, the final concentration of quercetin-3-O-glucoside (Q3G) was directly proportional to the rutin concentration and inversely proportional to the reaction time. At an anti-proliferative concentration (2.5 μg/mL), hydrolyzed rutin derivatives did not show a mutagenic effect, except for the sample with a higher content of Q3G (after 4 h of the enzymatic hydrolysis of rutin). Moreover, the higher Q3G content in hydrolyzed rutin protected the CHO-K1 cells 92% of the time against methyl methanesulfonate-induced mutagenic damage. These results suggested that the anti-mutagenic effect of hydrolyzed rutin might be related to antioxidant and cell death induction. Presenting a good lipophilicity/hydrophilicity ratio, together with antioxidant and anti-mutagenic activities, the hesperidinase-mediated hydrolyzed rutin seemed to be a promisor raw material for the development of food supplements.

## 1. Introduction

In addition to nutrients, plant foods are a rich source of bioactive substances such as carotenoids and flavonoids. The frequent consumption of these substances has been associated with a reduced risk of developing chronic and degenerative diseases. Several of these diseases are triggered by genetic alterations mediated by exposure to genotoxic agents, such as UV radiation, chemical substances and some viruses [1,2,3].

Due their physicochemical characteristics, free flavonoids (aglycones) have low oral absorption. In comparison to aglycone forms, the presence of sugar moieties usually increases flavonoids’ bioavailability, glucosides absorbed more rapidly than rhamnosides and rhamnoglucosides. The presence of hydrolyzing enzymes, such as lactase phlorizin hydrolase and β-glucosidase, in the small intestine, together with Na^+^-dependent glucose transporter 1 (SGLT1), on epithelial cells can explain this evidence [4,5].

To improve the industrial use of rutinosides, such as rutin, different methods, including enzymatic hydrolysis, have been developed. Rutin (quercetin-3-O-rutinoside) can be enzymatically converted to quercetin-3-O-glucoside (Q3G or isoquercitrin) by the removal of the terminal rhamnose by different enzymes. Several biological effects, such as anti-inflammatory, atheroprotective, antioxidant and anti-proliferative activities, have been described for Q3G. The beneficial effects can also be attributed to the increased blood level of quercetin detected after Q3G consumption [4,5,6,7,8]. A previous study of our research group evaluated α-L-rhamnosidases (hesperidinase from *Penicillium* sp. and naringinase from *Penicillium decumbens*) for the hydrolysis of rutin to obtain Q3G, which displayed higher scavenging and anti-proliferative activities than rutin [9]. Exhibiting both α-l-rhamnosidase and β-d-glucosidase activities, the commercial hesperidinase catalyzes the hydrolysis of rutin, providing Q3G and quercetin (Figure 1). The previous thermal treatment of hesperidinase results in the inhibition of the activity of β-D-glucosidase decreasing Q3G metabolism [9].

Considering that the ability to scavenge reactive oxygen species and the induction of detoxification pathways have been highlighted as possible mechanisms of action evolved in the anti-mutagenic activity and chemopreventive effects of secondary metabolites [1,2,3], the present study aimed to evaluate the kinetic parameters of hesperidinase-mediated rutin hydrolysis, together with the in vitro evaluation of anti-proliferative and (anti)mutagenic activities in the CHO-K1 cell line.

## 2. Materials and Methods

### 2.1. Enzymes and Reagents

Hesperidinase from *Penicillium* sp. (catalog number H8510), rutin (95% min., catalog number R5143), quercetin-3-glucoside (catalog number 17793) and quercetin (catalog number Q4951) standards were purchased from the Sigma-Aldrich Chemical Co. All solvents and other reagents were of analytical, spectrometric or chromatographic grade.

### 2.2. Bioconversion Reaction

Hesperidinase solution (50 mg L^−1^ in 0.05 M acetate buffer pH 4.0) were heated at 70 °C for 30 min to inactivate glucosidase activity, as previously described [9]. According to the manufacturer’s information, hesperidinase expresses both α-l-rhamnosidase (EC 3.2.1.40) and β-d-glucosidase (EC 3.2.1.21) activities. One unit will liberate 1.0 µmol of reducing sugar (as glucose) from hesperidin per min at pH 3.8 at 40 °C. The reaction mixture containing 100 µL of enzyme preparation (50 mg L^−1^) and 4 mL of rutin solution (previously dissolved in 1 mL of methanol) was mixed and incubated for 4, 8 and 12 h with shaking (130 rpm) at 40 °C. The reaction was stopped by boiling (100 °C) for 30 min. The mixture was subjected to preparative TLC for the isolation of Q3G, and the corresponding spot was scrapped, treated with methanol and filtered through Whatman filter paper, as previously described [10]. The procedure was repeated using several silica plates. The resulting methanol solution was evaporated, subsequently the freeze-dried and stored at −20 °C until the analysis.

### 2.3. Determination of Kinetic Parameters

In order to measure the Michaelis–Menten constant (Km) and the maximum reaction rate (Vmax), different substrate concentrations of rutin solution (2.5–10 mM) were tested in the assay system. Vmax and Km were calculated using Sigma Plot software (Aspire Software International, Ashburn, VA, USA). All samples were assayed in triplicate at 40 °C, 130 rpm min^−1^ in a shaker, and the aliquots were obtained at 4, 8 and 12 h of enzymatic reaction. To attain equilibrium, the samples kept at −20 °C until the analysis.

### 2.4. Mass Spectrometric Analysis by Ultra-Performance Liquid Chromatography/Time-of-Flight Mass Spectrometry (UHPLC/QTOF-MS^E^)

Samples were reconstituted with in methanol:DMSO 1:1 (*v/v*) to a final concentration of 10 µg mL^−1^ before the LC-MS analysis. Data were acquired using an ACQUITY FTN liquid chromatograph coupled to a XEVO-G2XS QTOF mass spectrometer (Waters, Manchester, NY, USA, Reino Unido) using MassLynx 4.1 software, as previously described [11]. Briefly, an Acquity UPLC^®^ CSH C18 (2.1 mm × 100 mm × 1.7 μm, Waters) column was used. The mobile phase consisted of water and formic acid (0.1%) (A) and acetonitrile (B) at a flow rate of 0.40 mL min^−1^ with a linear gradient (in % B): 0–8.0 min: 5%; 8.0–8.5 min: 95%; 8.5–8.6 min: 99% (with a further 1.6 min for column re-equilibration), resulting in a 10 min analysis method. The injection volumes were 0.5 and 2.0 µL for the positive and negative mode, respectively. The column oven was kept at 45 °C. For the electrospray ionization source, the parameters were set as follows for both the positive and negative mode: capillary voltage of 2.5 kV, sampling cone of 40,000, source temperature of 140 °C, desolvation temperature of 550 °C, cone gas flow of 50 L h^−1^ and desolvation gas flow of 900 L h^−1^. The acquisition scan range was from 100 to 1000 Da in the centroid, and the analyses used a data-independent acquisition (MS^E^) approach. Leucine encephalin (molecular weight = 555.62; 200 pg μL^−1^ in 1:1 ACN:H_2_O solution) was used as the lock mass for accurate mass measurements, and a 0.5 mM sodium formate solution was used for instrument calibration. For the quantification of Q3G, the standard stock solution was prepared by dissolving 1 mg of Q3G in 1 mL of methanol. Calibration curves (1–10 μM) were plotted with the peak area (*Y*-axis) versus standard concentrations (*X*-axis). The linear regression equation was y = 9373.1x + 153.1 (R^2^ = 0.9986).

### 2.5. Partition Coefficient Determination in Octanol/Water (k)

The partition coefficients in octanol/water (k) were analyzed to determine the lipophilicity of the samples before and after hydrolysis. In test tubes, 2.0 mL of a solution of each sample with a concentration of 50 µM were added to 2.0 mL of octanol saturated with water. The mixture was shaken for 1 min and then centrifuged (15 min, 3000 rpm). After filtering through a 0.22 µm polyethylene filter with a PTFE membrane (Merck Millipore, Billerica, MA, USA), the compound concentration was determined for each phase by UHPLC/QTOF-MS^E^. The partition coefficient was obtained using the equation: K = Co/Ca, where Co = the test compound concentration in octanol and Ca = the test compound concentration in an aqueous solution.

### 2.6. In Vitro Evaluations

#### 2.6.1. Cell Line

Immortalized Chinese Hamster ovary epithelial cells (CHO-K1) kindle provided by Dr. Mario S. Mantovani, State University of Londrina, was maintained in complete medium (RPMI 1640 (GIBCO^®^), supplemented with 5% of fetal bovine serum (FBS-Gibco^®^) and 1% of penicillin: streptomycin solution (1000 U ml^−1^: 1000 mg ml^−1^) (Vitrocell^®^)) at 37 °C in humid atmosphere with CO_2_ 5%. The experiment was performed with the cell line between passages 5 and 12.

#### 2.6.2. Samples Preparation

Aliquots of rutin; hydrolyzed rutins (4, 8, and 12 h reaction times); quercetin and methyl methanesulfonate (MMS, Sigma-Aldrich, Cod.78697) were diluted (1:10 *p/v*) in dimethyl sulfoxide (DMSO, Synth) before dilution in complete medium to afford the final concentrations described in each assay.

#### 2.6.3. Anti-Proliferative Activity Assay

The influence on cell growth of selected samples was evaluated following the protocol described by Monks et al. [12] with adaptations [13]. In 96-well plates, CHO-K1 cells (4 × 10^4^ cell mL^−1^, 100 µL well^−1^) were exposed to each sample (0.25, 2.5, 25 and 250 µg mL^−1^, final concentrations) for 4 and 48 h. In the 4 h exposure experiment, after the first 4 h, the medium was removed, and CHO-K1 cells were kept in fresh complete medium for 20 h. A T0 plate representing the cell density at the time of sample treatment was made (T0_untreated cells_). At each experimental plate (4 + 20 h and 48 h), untreated cells were kept in the complete medium (T1_untreated cells_). The cell viability was evaluated by total protein quantification at 540 nm with the sulforhodamine assay. For each sample, the influence of each concentration on cell growth was expressed in percentage, considering (T1_untreated cells_–T0_untreated cells_) as 100% cell growth. The sample concentration required by eliciting 50% cell growth inhibition (GI_50_) was calculated by sigmoidal regression.

#### 2.6.4. Cytokinesis-Block Micronucleus (CBMN) Assay

The experiments were conducted according to the guideline OECD 487 [14] with few modifications [15].

For the mutagenic effect evaluation, the CHO-K1 cells (2 × 10^5^ cells well^−1^) in 6-well plates were exposed to rutin; hydrolyzed rutins (4, 8, and 12 h reaction times) and quercetin (2.5 µg ml^−1^, in duplicate) for 4 h. Untreated cells, as the negative control, DMSO-treated (0.25%) as the solvent control and MMS-treated (25 µg ml^−1^) as the micronuclei inductor control cells were prepared. After cell washing (phosphate buffer, pH = 7.0, 2 mL well^−1^), cytochalasin B solution (3 µg ml^−1^, 2 mL well^−1^) was added. After 20 h, cells were detached, fixed with sodium citrate solution (1% *w/v*), followed by methanol:acetic acid 3:1 (*v/v*) (three times) and dropped into slides (3 slides/cell suspension). All slides were maintained at 65 °C in a wet atmosphere for 3 min before being stained with 5% Giemsa solution (Dinâmica^®^) for 20 min, washed with distilled water and dried at room temperature.

For the anti-mutagenic assay, the CHO-K1 cells (2 × 10^5^ cells well^−1^) in 6-well plates were exposed for 4 h to rutin, hydrolyzed rutin (8 h reaction time), quercetin (2.5 µg ml^−1^, in duplicate) and MMS (25 µg ml^−1^). After 4 h exposure, cells were washed (phosphate buffer, pH = 7.0, 2 mL well^−1^), exposed to cytochalasin B solution (3 µg ml^−1^, 2 mL well^−1^) for 20 h and the slides were prepared as explained for mutagenic evaluation.

For each experiment, at least 2000 binucleated (BN) cells, besides mononucleated (MoN) and multinucleated cells (MuN), were counted (1000 binucleated cells/replicate) under 400× magnification using a light microscope (Leica, SME model). In each cell population, the micronuclei frequency (CBMN, Equation (1)) was calculated as:CBMN (%) = [(MN)/(BN)] × 100(1)
where MN was the number of binucleated cells with 1, 2 or 3 micronuclei and BN was total number of binucleated cells.

For the evaluation of the cytotoxic effects, two parameters, named the Cytokinesis-Block Proliferation Index (CBPI, Equation (2)) and Replication Index (RI, Equation (3)), were calculated as:CBPI = (1 × MoN + 2 × BN + 3 × MuN)/(MoN + BN + MuN)(2)
RI (%) = [(BN + 2 × MuN)/(BN + MuN)]control/[(BN + 2 × MuN)/(BN + MuN)] × 100(3)

### 2.7. Statistical Analysis

The data were expressed as means ± the standard deviation (SD). The statistical significance of the analytical results was assessed by one-way ANOVA, and the differences identified were pinpointed by an unpaired Student’s *t*-test for the enzymatic and chromatographic assays. For the in vitro evaluation, the significance of the observed differences was evaluated by one-way ANOVA, followed by Tukey’s test. An associated probability (*p* value) of less than 5% was considered significant.

## 3. Results

### 3.1. Hydrolysis Kinetics of Rutin

Up to 9 mM, the final concentration of Q3G was directly proportional to the rutin concentration and inversely proportional to the reaction time (Figure 2) during the hesperidinase-catalyzed conversion. Based on these experimental results, the Michaelis–Menten constant (Km) and the maximum reaction rate (Vmax) parameters were calculated (Table 1).

The maximum rate of Q3G production was 6.54 mM at 4 h of enzymatic reaction, with an apparent kinetic parameter Vmax/Km of 35.28 × 10^−2^ h^−1^. An increasing reaction time (8 and 12 h) resulted in a decrease in reaction efficiency. The higher concentration of Q3G after 4 h of reaction resulted in increased hydrophilicity, as observed by the partition coefficient k (Table 1).

### 3.2. Analysis of the Reaction Products by UHPLC/QTOF-MS^E^

The UPLC-MS analysis of hydrolyzed rutin after 4 h hesperidinase reaction (Figure 3a) resulted in 64.7% of quercetin-3-glucoside (retention time rt = 4.69 min), 8.3% of quercetin (rt = 5.91 min) and 27% of residual rutin (rt = 3.71 min). After 8 and 12 h of reaction, Q3G was partially converted to quercetin, as evidenced by an increased amount of quercetin (≈13%) (Figure 3b). The purity of isolated Q3G checked by UPLC was found to be 78.5%. A representative MS^E^ spectrum, in the negative ion mode, showed the peaks at *m/z* 609.14, 463.10 and 301.08 attributed to rutin, Q3G (loss of the rhamnose group from rutin) and quercetin (loss of the glucose group from Q3G or loss of the rhamnosil-glucose group from rutin), respectively (Figure 3c).

### 3.3. In Vitro Toxicological Evaluation

Hydrolyzed rutin samples showed a cytostatic effect on CHO-K1 cells similar to that observed for quercetin, independent of the reaction time (Figure 4, Table 2). Increasing the exposure time (from 4 + 20 to 48 h) resulted in reduced anti-proliferative activity against CHO-K1 cells for almost all samples. Based on these results, the mutagenic and anti-mutagenic evaluations of rutin, hydrolyzed rutin samples and quercetin were done at 2.5 µg ml^−1^.

At the selected experimental conditions in the mutagenic assay, no sample affected CHO-K1 cell proliferation during the Cytokinesis-Block Micronucleus (CBMN) assay, affording a proliferation index (CBPI) higher than 1.5 and replication index (RI) higher than 80%. Methyl methanesulfonate (MMS) increased in an almost 5× micronuclei (MN) frequency compared to the untreated cells. Both rutin and quercetin did not induce an increase in MN frequency, while only hydrolyzed rutin 01 (after 4 h of reaction) slightly increased MN frequency (Table 3).

Rutin, quercetin and the hydrolyzed rutin 01 (after 4 h of reaction) were evaluated as anti-mutagenic agents in the Cytokinesis-Block Micronucleus (CBMN) assay. No sample affected cell proliferation, affording a proliferation index (CBPI) higher than 1.6 and replication index (RI) higher than 75%, at the experimental conditions. All samples showed an anti-mutagenic effect protecting (82–92%) CHO-K1 cells from MMS-induced mutagenesis (Table 4).

## 4. Discussion

Continuing the study on the enzymatic hydrolysis of rutin [9], this paper described some kinetics aspects of hesperidinase-mediated reaction together with the (anti)mutagenic evaluation of the hydrolyzed products.

Rutin was efficiently converted to intermediate Q3G by the rhamnosidase activity of hesperidinase (hydrolysis of the rhamnose group), with little quercetin produced after 4 h of enzymatic treatment. The apparent time-dependent reduction in the hydrolysis efficiency reflected the lower concentration of Q3G remaining in the system, as previously described [9]. After 8 and 12 h of enzymatic reaction, the reduction in the Q3G concentration could be attributed to the residual glucosidase activity of hesperidinase generating the aglycone quercetin (loss of glucose group from Q3G, Figure 1). The thermal inactivation process of undesirable b-D-glucosidase activity of hesperedinase, as described in the previous work [9], was not sufficient to prevent the hydrolysis of Q3G in longer reaction times.

The hydrophilic/lipophilic properties of rutin (substrate), quercetin (standard) and its derivative formed after the hydrolysis of rutin at different reaction times (4, 8 and 12 h) were compared (Table 1). By relating single-solute partitions between polar (water) and nonpolar (octanol) phases, the partition coefficient allows to infer some pharmacokinetic parameters, such as interphase partition, protein and membrane interactions, transport, absorption and excretion [16].

Flavonoids are usually poorly soluble in aqueous solutions, and the effect is dependent on the type and number of sugar moieties [4,5,6]. As expected, quercetin showed the highest partition coefficient, reflecting a higher degree of lipophilicity, while the partition coefficient observed for rutin indicated a higher degree of hydrophilicity. The hydrolyzed rutin showed partition coefficients between 0.64 and 0.80, reflecting the higher water solubility of Q3G in comparison to rutin and quercetin. These results were in agreement with those described in the literature [17,18,19]. Together with being slight less soluble in water, flavonoids bound to rhamnosyl need to be hydrolyzed by the colonic microbiota to allow aglycone release and, hence, absorption, while glucoside derivatives can be absorbed in the small intestine [4,5,6,18]. In this context, the enzymatic hydrolysis of rhamnosyl derivatives, such as rutin, is a good strategy to increase the bioavailability.

The cytostatic effect of rutin, quercetin and hydrolyzed rutin samples was evaluated following the NCI protocol [12] for anti-proliferative activity. In this protocol, untreated cells were evaluated at before (T0) and after (T1) the exposure time, allowing to distinguish the cytostatic and cytocidal effects. The exposure time (4 + 20 h) was selected, because it was the exposure time used in the CBMN assay [15], while the second one (48 h) was the same used in previous study against human tumor cell lines [9]. At these conditions, hydrolyzed samples with higher quercetin amounts (after 8 and 12 h of hydrolysis) showed an anti-proliferative effect similar to that observed for quercetin. The time exposure-dependent increase on effective concentrations observed for these samples may suggest some adaptive response from CHO-K1 cells. Further experiments should be done to elucidate this evidence. Moreover, this time-dependent variation on the anti-proliferative effect was not observed for hydrolyzed rutin 01 (64.7% Q3G, weak anti-proliferative effect) and rutin (inactive).

Previous results indicated that hesperidinase-mediated hydrolyzed rutin samples showed anti-proliferative activity against some tumor cell lines, together with an antioxidant effect in different chemical models [9]. In the present study, the mutagenic and anti-mutagenic effects of these samples were evaluated. The use of cytokinesis blockage allows the evaluation of micronuclei frequency (CBMN) in binucleate cells, providing high accuracy in the mutagenic analysis. However, it is important that the higher sample concentration did not affect cell proliferation. To assure this, the OECD 487 guidelines propose two parameters, named cytokinesis-block proliferation (CBPI) and replication (RI) indexes, to indicate cell proliferation and the number of cell cycles, respectively, during the time exposure to cytochalasin B [14]. It was established that CBPI values higher than 1.5 and RPI values higher than 55% were indicative of a low cytostatic effect of the sample under evaluation [14,20,21]. Considering the results described in Table 3 and Table 4, the CBPI and RI values observed for MMS (lower in the first experiment in comparison to the second) may explain why MMS, at the same concentration, induced different rates of increase in CBMN. Frequently used as a cryoprotective agent and solvent in biological studies [22,23], DMSO is reported as a non-mutagenic in vitro (up to 5000 µg/plate in the Ames assay) and in vivo (up to 700 mg/kg in Sprague–Dawley rats) [23]. In immortalized odontoblast-like MDPC-23 cells, DMSO, at 0.1 mM, induced a slight increase in cell viability [24], as observed in the present study.

In the present study, almost all samples showed non-mutagenic effects. The hydrolyzed rutin 01 promoted a slight increase in CBMN compared to the untreated cells. However, this difference was not statistically significant. Selected by its desirable characteristics (high Q3G content, antioxidant activity and partition coefficient), the hydrolyzed rutin 01 protected CHO-K1 cells from the MMS-induced mutagenic effect similar to quercetin and rutin. Although it seems contradictory, mutagenic effects that result in the activation of regulated cell death mechanisms have been described as a mechanism of anti-mutagenic activity of polyphenols [25], along with the ROS scavenging ability and activation of DNA repair mechanisms [1,2]. According to the literature, the mutagenic and anti-mutagenic effects of rutin, Q3G and quercetin have already been demonstrated in different experimental models. Despite in vitro mutagenic effects, these flavonoids seemed to be non-genotoxic in the in vivo evaluations (Table 5).

## 5. Conclusions

The use of heat-treated hesperidinase in the hydrolysis of rutin seems to be a promisor strategy to produce, in a short-term reaction (4 h), a hydrolyzed rutin containing high amount of Q3G and presenting good lipophilicity/hydrophilicity ratio, along with anti-mutagenic activity.

## Figures and Tables

**Figure 1 life-13-00549-f001:**
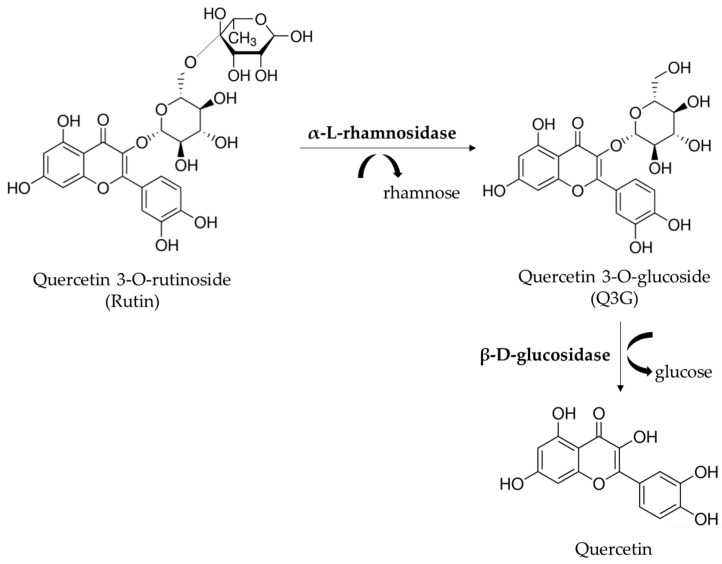
Hydrolysis of rutin mediated by commercial hesperidinase. The commercial hesperidinase from *Penicillium* sp. acts both as α-*l*-rhamnosidase and β-*d*-glucosidase, affording quercetin 3-O-glusoside (Q3G) and quercetin, respectively.

**Figure 2 life-13-00549-f002:**
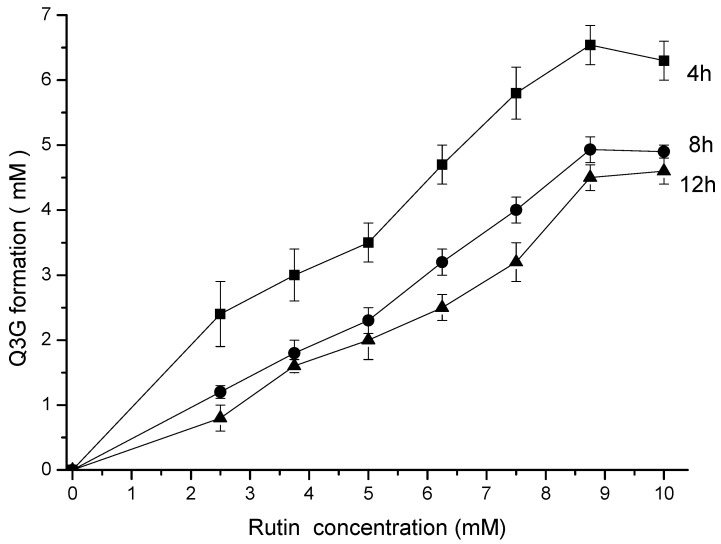
Kinetics of the enzymatic hydrolysis of rutin and Q3G formation. The incubation time was 4 (■), 8 (●) and 12 (▲) h.

**Figure 3 life-13-00549-f003:**
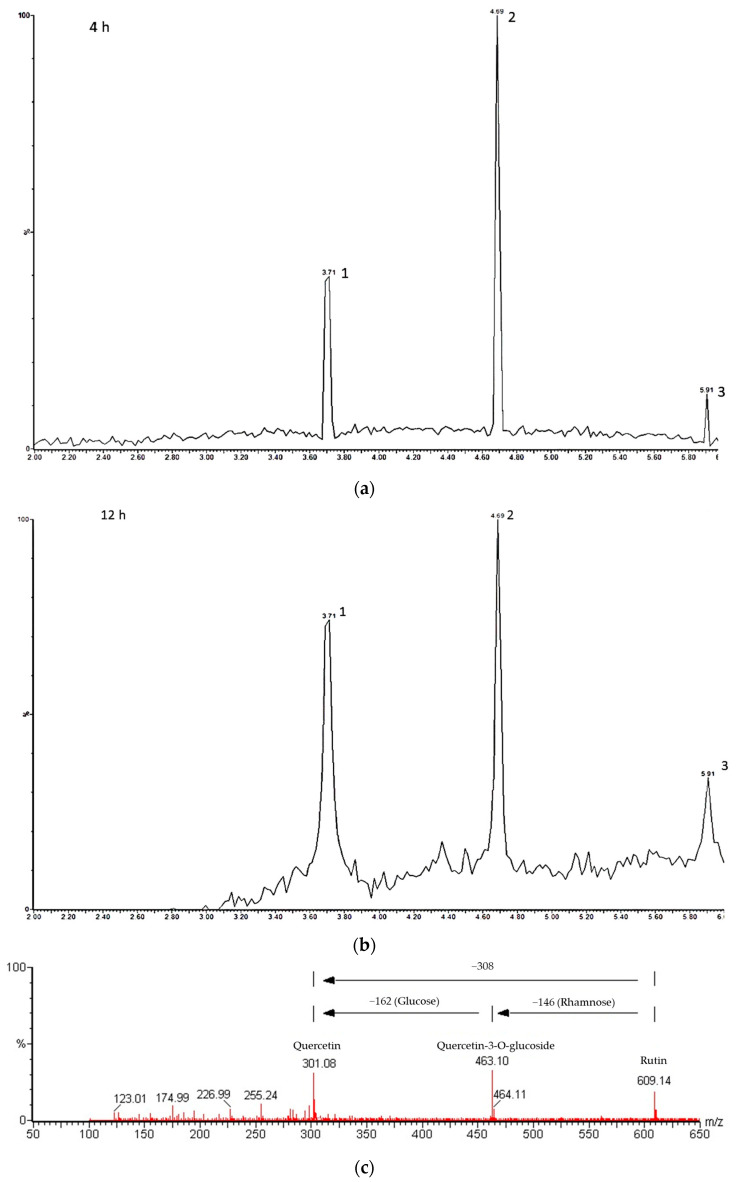
UPLC-MS analysis of hydrolyzed rutin. Chromatograms of hydrolyzed rutin after 4 h (**a**) and 12 h (**b**) of hesperidinase treatment; peak 1: rutin, peak 2: quercetin-3-glucoside and peak 3: quercetin; (**c**) representative mass spectrum acquired in negative ion mode.

**Figure 4 life-13-00549-f004:**
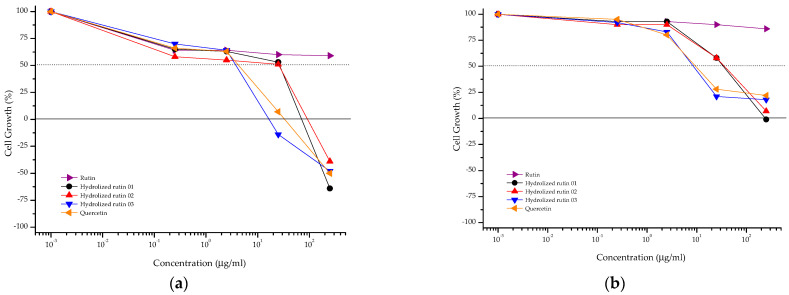
Anti-proliferative profile of rutin, hydrolyzed rutin 01 (after 4 h of reaction), hydrolyzed rutin 02 (after 8 h of reaction), hydrolyzed rutin 03 (after 12 h of reaction) and quercetin against the CHO-K1 cell line after 4 + 20 h (**a**) or 48 h (**b**) of exposure. Sample concentration range: 0.25 to 250 μg/mL.

**Table 1 life-13-00549-t001:** Kinetics of the enzymatic hydrolysis of rutin and the partition coefficient in octanol/water (k) of the derivative formed after 4, 8 and 12 h of reaction.

Samples		k ^1^	Q3G (mM)	Vmax (mM h^−1^)	Km (mM) ^2^	Vmax/Km (×10^−2^ h^−1^)
Hydrolyzed rutin	01	0.64	6.54 ± 0.25	1.63	4.62	35.28
	02	0.78	4.93 ± 0.16	0.62	5.36	11.57
	03	0.80	4.60 ± 0.21	0.38	5.72	6.64
Rutin		0.89	-	-	-	-
Quercetin		1.59	-	-	-	-

^1^ Standard deviation < 0.01; ^2^ standard deviation < 0.03; k = partition coefficient. Samples: Hydrolyzed rutin 01, 02 and 03 = enzymatic product after 4, 8 and 12 h, respectively; Rutin = substrate; Quercetin = standard.

**Table 2 life-13-00549-t002:** Anti-proliferative effect of rutin, hydrolyzed rutins and quercetin against CHO-K1 cells expressed as concentrations required to inhibit 50% of cell growth (GI_50_, μg/mL).

Sample ^1^		GI_50_ (μg/mL)	
		4 + 20 h ^2^	48 h ^2^
Hydrolyzed rutin	01	25.0 ± 0.9	29.6 ± 7.8
	02	2.5 *	31.1 ± 10.7
	03	2.5 *	9.1 ± 3.4
Rutin		>250	>250
Quercetin		2.5 *	11.8 ± 5.0

Results expressed as the mean ± standard error of technical duplicates of one experiment. ^1^ Sample: Hydrolyzed rutin 01, 02 and 03 = enzymatic product after 4, 8 and 12 h of reaction, respectively; Rutin (substrate); Quercetin (standard). ^2^ Exposure time: 4 + 20 h = CHO-K1 cells exposed during 4 h to each sample, followed by 20 h of recovery; 48 h = CHO-K1 cells exposed during 48 h to each sample. * Approximated GI_50_: standard error higher than calculated concentration.

**Table 3 life-13-00549-t003:** In Vitro mutagenic effect of rutin, hydrolyzed rutins and quercetin.

Control/Sample ^1^		Parameters ^2^		
		CBPI	RI	CBMN
CHO-K1		1.7 ± 0.0 ^a,c^	100.0 ± 0.0 ^a^	1.05 ± 0.06 ^a,c^
MMS		1.58 ± 0.02 ^b^	88.5 ± 2.7 ^a^	4.6 ± 0.4 ^b^
DMSO		1.89 ± 0.03 ^c^	137.5 ± 4.5 ^b^	0.89 ± 0.09 ^a^
Hydrolyzed rutin	01	1.69 ± 0.06 ^a,b^	105.4 ± 9.1 ^a^	2.9 ± 0.2 ^b,c^
	02	1.66 ± 0.06 ^a,b^	101.8 ± 9.3 ^a^	1.4 ± 0.2 ^a,c^
	03	1.645 ± 0.007 ^a,b^	99.7 ± 0.8 ^a^	1.6 ± 0.4 ^a,c^
Rutin		1.68 ± 0.02 ^a,b^	103.5 ± 3.3 ^a^	2.3 ± 0.2 ^a,c^
Quercetin		1.65 ± 0.05 ^a,b^	99.8 ± 7.3 ^a^	2.0 ± 1.2 ^a,c^

Results expressed as the mean ± standard deviation of technical duplicates of one experiment. ^1^ Controls: CHO-K1 = untreated cells (negative control); MMS = methanesulfonate (25 μg/mL, positive control); DMSO = dimethyl sulfoxide (0.25%, solvent control); Samples (2.5 μg/mL) = Hydrolyzed rutin 01, 02 and 03 (enzymatic product after 4, 8 and 12 h of reaction, respectively); Rutin (substrate); Quercetin (standard). ^2^ Parameters: CBPI = Cytokinesis-Block Proliferation Index; RI = Replication index; CBMN = Cytokinesis-Block Micronuclei frequency. Statistical analysis: one-way ANOVA, followed by Tukey’s test (different letters in the same column represent significant differences, *p* ≤ 0.001).

**Table 4 life-13-00549-t004:** In Vitro anti-mutagenic effect of rutin, hydrolyzed rutin and quercetin.

Control/Sample ^1^		Parameters ^2^		
		CBPI	RI	CBMN
CHO-K1		1.82 ± 0.02 ^a^	100.0 ± 0.0 ^a^	1.395 ± 0.007 ^a^
MMS		1.8 ± 0.0 ^a^	100.1± 2.7 ^a^	8.0 ± 0.5 ^b^
DMSO		1.86 ± 0.03 ^a^	104.0 ± 6.3 ^a^	1.4 ± 0.2 ^a^
MMS +	Hydrolyzed rutin 01	1.65 ± 0.04 ^b^	78.3 ± 3.2 ^b^	0.7 ± 0.3 ^a^
	Rutin	1.66 ± 0.04 ^b^	79.11 ± 1.9 ^b^	0.9 ± 0.4 ^a^
	Quercetin	1.62 ± 0.03 ^b^	75.4 ± 5.5 ^b^	1.4 ± 0.3 ^a^

Results expressed as the mean ± standard deviation of technical duplicates of one experiment. ^1^ Control: CHO-K1 = untreated cells (negative control); MMS = methyl methanesulfonate (25 μg/mL, positive control); DMSO = dimethyl sulfoxide (0.25%, solvent control); Sample: Hydrolyzed rutin 01= after 4 h of the enzymatic hydrolysis of rutin (2.5 μg/mL); Rutin = substrate (2.5 μg/mL); Quercetin = standard (2.5 μg/mL). ^2^ Parameters: CBPI = Cytokinesis-Block Proliferation Index; RI = Replication index; CBMN = Cytokinesis-Block Micronuclei frequency. Statistical analysis: one-way ANOVA, followed by Tukey’s test (different letters in the same column represent significant differences, *p* ≤ 0.001).

**Table 5 life-13-00549-t005:** Some reported evidence on the mutagenic and anti-mutagenic effects of rutin, Q3G and quercetin in preclinical models.

Sample	Model	Treatment	Effect	Ref.
Quercetin	Comet assay in HepG2 Cells	Up to 10.0 μg/mL	No genotoxic effect ↓ B[a]P, MMS or DXR-induced DNA damage	[26]
	SCE and MN frequency in CHO cells	Up to 30 μM	↑ SCE (at 0.3 μM) and ↑ MN frequency (at 30 μM)	[27]
	Ames test (*S. typhimurium* TA100, TA98 and TA102)	1.0 and 0.3 μM	↑ MN frequency ↓ H_2_O_2_-induced oxidative damage	[28]
	MN frequency in mouse bone marrow erythrocytes	up to 10 mg/kg, v.o., 1–5 day-treatment	↑ MN frequency ↓ B[a]P-induced MN frequency	[29]
	MN frequency in mouse bone marrow erythrocytes (female and male)	Up to 558 mg/kg, i.p., single dose	No mutagenic effect	[30]
	MN frequency in mice bone marrow (female and male)	Up to 2500 mg/kg, v.o., 2 day-treatment	No mutagenic effect	[31]
	Comet assay in mice bone marrow (female and male)		↑ DNA damage, dose-independent	
	MN frequency in bone marrow in rats	Up to 2000 mg/kg, v.o., single dose	No mutagenic effect	[32]
Q3G	SCE and MN frequency in CHO cells	Up to 2 mM	↑ SCE (at 2 mM) and ↑ MN frequency (at 2 mM)	[27]
	Ames test (*S. typhimurium* TA100, TA98 and TA102)	0.1–2.2 μM	Negligible mutagenic effect ↓ H_2_O_2_-induced oxidative damage	[28]
	Ames test (*S. typhimurium* TA100, TA98, TA1535, and TA1537)	Up to 5000 μg/mL	Dose-dependent mutagenic effect, with and without metabolic activation	[33]
	MN frequency in human TP53 competent TK6 cells	Up to 1000 μg/mL	No mutagenic effect, with and without metabolic activation	
	Chromosomal aberration assay in CHO-WBL cells	Up to 1500 μg/ml	No mutagenic effect, with metabolic activation	
	MN frequency in peripheral blood in rats	Up to 2000 mg/kg, v.o., 3 day-treatment	No mutagenic effect	
Rutin	Comet assay in HepG2 Cells	Up to 50.0 μg/mL	No genotoxic effect ↓ B[a]P, MMS or DXR-induced DNA damage	[26]
	Ames test (*S. typhimurium* TA100, TA98 and TA102)	0.1–1.6 μM	No mutagenic effect ↓ H_2_O_2_-induced oxidative damage	[28]
	SCE and MN frequency in CHO cells	Up to 2 mM	↑ SCE (at 10 μM) and ↑ MN frequency (at 2 mM)	[27]
	MN frequency in mice bone marrow (female and male)	Up to 2500 mg/kg, v.o., 2 day-treatment	No mutagenic effect	[31]
	Comet assay in mice bone marrow (female and male)		↑ DNA damage at 1250 mg/kg	

B[a]P = Benzo[a]pyrene; MMS = methylmethanesulphonate; DXR = doxorubicin; SCE = sister chromatid exchanges; MN = micronuclei; ↓ = damage reduction, ↑ = damage increase (damage: SCE, MN, DNA fragmentation, or oxidative damage).

## Data Availability

The dataset analyzed during the current study is available from the corresponding author Ana Lucia T. G. Ruiz on reasonable request.

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
