# Peer review of "Enzymatic Hydrolysis of Rutin: Evaluation of Kinetic Parameters and Anti-Proliferative, Mutagenic and Anti-Mutagenic Effects"

_life, 2023, doi:10.3390/life13020549_

Round 1

Reviewer 1 Report

The article “Enzymatic hydrolysis of rutin: evaluation of kinetic parameters and anti-proliferative, mutagenic and anti-mutagenic effects” described the kinetics of hesperidinase-mediated hydrolysis of rutin to Q3G and biological evaluation of the products. There are some points for improvement and discussion.

1.       In the Introduction, an enzymatic reaction and the chemical structures of rutin, Q3G, and rutin should be added for the reader’s better understanding.

2.       In the discussion (lines 282-285), the author suggested only that "the apparent time-dependent reduction in the hydrolysis efficiency reflected the lower concentration of rutin remaining in the system, as previously described." Is it possible that at a later time than 4 h, Q3G was converted to the other compound, and the second reaction caused a lower concentration of Q3G?

3.       The partition coefficient values of samples 01-03 in Table 1 revealed that the longer the hydrolysis time, the greater the lipophilic products (perhaps quercetin?). To determine whether the amount of quercetin increased over time, UHPLC chromatograms of the products at 8 and 12 hours should be provided. 

Reviewer 2 Report

The topic is positively and will be interesting to the readers of the journal. Overall, the structure is positive, and the results are technically sound. However, the presentation of the results, statistics, and discussion is not so clear in the manuscript. It is impossible to highlight the value of this article. I would encourage the authors to consider the comments listed below in order to clarify the information provided.

 1.          Increasing the reaction time increases the number of effective collisions of the reactant molecules. Please explain why, under the same reaction conditions, the increase in reaction time (8 and 12 hours) decreases the efficiency of the reaction.

2.          Please explain, why in the anti-proliferation experiment, the exposure time was 4 hours or 48 hours? Furthermore, in the GI50 concentration, why is the cellular inhibitory concentration at 4 hours of exposure relatively lower than 48 hours.  Please add clarification to the Discussion.

3.          In figure 3, Please specify the concentration used for the anti-proliferative assay.

4.          Methyl methanesulfonate (mms) is used as a positive control group, why are there different results in Table 4 and Table 5. The results of Cytokinesis-Block Proliferation Index (CBPI) and Replication index (RI) under MMS treatment were the same as those of untreated cells.

5.          In addition, it can be explained that the replication index (RI) higher than 75% has no effect on cell proliferation, please write the reference?

6.          Please explain in Table 4 why treatment of cells with DMSO significantly increases CBPI?

7.          Hydrolyzed rutin 01 in Table 4 mutagenic experiments, it seems to increase the results of cell mutagenesis. Please explain why only Hydrolyzed rutin 01 is used in the antimutagenic experiment in Table 5.  Please add clarification to the Discussion.

8.          In the article “in vitro” should use italics. Example: Line 67, 193, 254, 270, 321.

9.          Line 158, please clearly explain what are T1 untreated cells and T0 untreated cells.

Reviewer 3 Report

τ

Round 2

Reviewer 1 Report

1. Line 73-74 , the phrase “ increasing the efficiency of the conversion of rutin into Q3GW might be changed to “decreasing Q3G metabolism”.

2. Typos and spelling errors must be re-checked.

3. Please improve clarity of Figure 3a.

Author Response

Reviewer 01:

1. Line 73-74 , the phrase “ increasing the efficiency of the conversion of rutin into Q3GW might be changed to “decreasing Q3G metabolism”.

We thank the reviewer for the observation. The phrase was changed.

2. Typos and spelling errors must be re-checked.

We thank the reviewer for the observation. We carefully proofread the manuscript to correct typos and spelling errors.

3. Please improve clarity of Figure 3a.

We thank the reviewer for the observation. A new version of Figure 3 has been included to improve resolution.

Reviewer 2 Report

I have carefully reviewed all chapter content. I believe the author has spent a lot of time revising and providing a complete response. It must be affirmed, and I sincerely look forward to accepting and passing the publication of this article.

Best regards,

Dr. Chang

Author Response

Reviewer2: I have carefully reviewed all chapter content. I believe the author has spent a lot of time revising and providing a complete response. It must be affirmed, and I sincerely look forward to accepting and passing the publication of this article.

AA: We thank the reviewer for the care and attention in evaluating our manuscript.

Reviewer 3 Report

I appreciate the effort of the authors to correct or change and improve their paper according to the comments of the reviewers. In my opinion the article can be accepted in present form.

Author Response

Reviewer 3: I appreciate the effort of the authors to correct or change and improve their paper according to the comments of the reviewers. In my opinion the article can be accepted in present form.

AA: We thank the reviewer for the care and attention in evaluating our manuscript.